# Fitting Low-Rank Tensors in Constant Time

**Kohei Hayashi**[*]
National Institute of Advanced Industrial Science and Technology
RIKEN AIP
hayashi.kohei@gmail.com

**Yuichi Yoshida**[†]
National Institute of Informatics
yyoshida@nii.ac.jp

## Abstract

In this paper, we develop an algorithm that approximates the residual error of Tucker decomposition, one of the most popular tensor decomposition methods, with a provable guarantee. Given an order-$K$ tensor $X \in \mathbb{R}^{N_1 \times \cdots \times N_K}$, our algorithm randomly samples a constant number $s$ of indices for each mode and creates a "mini" tensor $\tilde{X} \in \mathbb{R}^{s \times \cdots \times s}$, whose elements are given by the intersection of the sampled indices on $X$. Then, we show that the residual error of the Tucker decomposition of $\tilde{X}$ is sufficiently close to that of $X$ with high probability. This result implies that we can figure out how much we can fit a low-rank tensor to $X$ *in constant time*, regardless of the size of $X$. This is useful for guessing the favorable rank of Tucker decomposition. Finally, we demonstrate how the sampling method works quickly and accurately using multiple real datasets.

## 1 Introduction

Tensor decomposition is a fundamental tool for dealing with array-structured data. Using tensor decomposition, a tensor (or a multidimensional array) is approximated with multiple tensors in lower-dimensional space using a multilinear operation. This drastically reduces disk and memory usage. We say that a tensor is of *order $K$* if it is a $K$-dimensional array; each dimension is called a *mode* in tensor terminology.

Among the many existing tensor decomposition methods, *Tucker decomposition* [18] is a popular choice. To some extent, Tucker decomposition is analogous to singular-value decomposition (SVD): as SVD decomposes a matrix into left and right singular vectors that interact via singular values, Tucker decomposition of an order-$K$ tensor consists of $K$ factor matrices that interact via the so-called core tensor. The key difference between SVD and Tucker decomposition is that, with the latter, the core tensor need not be diagonal and its "rank" can differ for each mode $k = 1, \ldots, K$. In this paper, we refer to the size of the core tensor, which is a $K$-tuple, as the *Tucker rank* of a Tucker decomposition.

We are usually interested in obtaining factor matrices and a core tensor to minimize the *residual error*— the error between the input and low-rank approximated tensors. Sometimes, however, knowing the residual error itself is an important task. The residual error tells us how the low-rank approximation is suitable to the input tensor, and is particularly useful to predetermine the Tucker rank. In real

---

[*]Supported by ONR N62909-17-1-2138.
[†]Supported by JSPS KAKENHI Grant Number JP17H04676 and JST ERATO Grant Number JPMJER1305, Japan.

---

**Algorithm 1**

---

**Input:** Random access to a tensor $X \in \mathbb{R}^{N_1 \times \cdots \times N_K}$, Tucker rank $(R_1, \ldots, R_k)$, and $\epsilon, \delta \in (0, 1)$.
  **for** $k = 1$ to $K$ **do**
      $S_k \leftarrow$ a sequence of $s = s(\epsilon, \delta)$ indices uniformly and independently sampled from $[N_k]$.
  Construct a mini-tensor $X|_{S_1, \ldots, S_K}$.
  Return $\ell_{R_1, \ldots, R_K}(X|_{S_1, \ldots, S_K})$.

---

applications, Tucker ranks are not explicitly given, and we must select them by considering the balance between space usage and approximation accuracy. For example, if the selected Tucker rank is too small, we risk losing essential information in the input tensor. On the other hand, if the selected Tucker rank is too large, the computational cost of computing the Tucker decomposition (even if we allow for approximation methods) increases considerably along with space usage. As with the case of the matrix rank, one might think that a reasonably good Tucker rank can be found using a grid search. Unfortunately, grid searches for Tucker ranks are challenging because, for an order-$K$ tensor, the Tucker rank consists of $K$ free parameters and the search space grows exponentially in $K$. Hence, we want to evaluate each grid point as quickly as possible.

Unfortunately, although several practical algorithms have been proposed, such as the higher-order orthogonal iteration (HOOI) [7], they are not sufficiently scalable. For each mode, HOOI iteratively applies SVD to an unfolded tensor—a matrix that is reshaped from the input tensor. Given an $N_1 \times \cdots \times N_K$ tensor, the computational cost is hence $O(K \max_k N_k \cdot \prod_k N_k)$, which depends crucially on the input size $N_1, \ldots, N_K$. Although there are several approximation algorithms [8, 21, 17], their computational costs are still intensive. Consequently, we cannot search for good Tucker ranks. Rather, we can only check a few candidates.

### 1.1 Our Contributions

When finding a good Tucker rank with a grid search, we need only the residual error. More specifically, given an order-$K$ tensor $X \in \mathbb{R}^{N_1 \times \cdots \times N_K}$ and integers $R_k \leq N_k$ $(k = 1, \ldots, K)$, we consider the following *rank-$(R_1, \ldots, R_K)$ Tucker-fitting* problem. For an integer $n \in \mathbb{N}$, let $[n]$ denote the set $\{1, 2, \ldots, n\}$. Then, we want to compute the following normalized residual error:

$$\ell_{R_1, \ldots, R_K}(X) := \min_{G \in \mathbb{R}^{R_1 \times \cdots \times R_K}, \{U^{(k)} \in \mathbb{R}^{N_k \times R_k}\}_{k \in [K]}} \frac{\left\| X - [\![ G; U^{(1)}, \ldots, U^{(K)} ]\!] \right\|_F^2}{\prod_{k \in [K]} N_k}, \quad (1)$$

where $[\![ G; U^{(1)}, \ldots, U^{(K)} ]\!] \in \mathbb{R}^{N_1 \times \cdots \times N_K}$ is an order-$K$ tensor, defined as

$$[\![ G; U^{(1)}, \ldots, U^{(K)} ]\!]_{i_1 \cdots i_K} = \sum_{r_1 \in [R_1], \ldots, r_K \in [R_K]} G_{r_1 \cdots r_K} \prod_{k \in [K]} U^{(k)}_{i_k r_k}$$

for every $i_1 \in [N_1], \ldots, i_K \in [N_K]$. Here, $G$ is the core tensor, and $U^{(1)}, \ldots, U^{(K)}$ are the factor matrices. Note that we are not concerned with computing the minimizer. Rather, we only want to compute the minimum value. In addition, we do not need the exact minimum. Indeed, a rough estimate still helps to narrow down promising rank candidates. The question here is how quickly we can compute the normalized residual error $\ell_{R_1, \ldots, R_K}(X)$ with moderate accuracy.

We shed light on this question by considering a simple sampling-based algorithm. Given an order-$K$ tensor $X \in \mathbb{R}^{N_1 \times \cdots \times N_K}$, Tucker rank $(R_1, \ldots, R_K)$, and sample size $s \in \mathbb{N}$, we sample a sequence of indices $S_k = (x_1^k, \ldots, x_s^k)$ uniformly and independently from $\{1, \ldots, N_k\}$ for each mode $k \in [K]$. Then, we construct a mini-tensor $X|_{S_1, \ldots, S_K} \in \mathbb{R}^{s \times \cdots \times s}$ such that $(X|_{S_1, \ldots, S_K})_{i_1, \ldots, i_K} = X_{x_{i_1}^1, \ldots, x_{i_K}^K}$. Finally, we compute $\ell_{R_1, \ldots, R_K}(X|_{S_1, \ldots, S_K})$ using a solver, such as HOOI, that then outputs the obtained value. The details are provided in Algorithm 1.

In this paper, we show that Algorithm 1 achieves our ultimate goal: with a provable guarantee, the time complexity remains *constant*. Assume each rank parameter $R_k$ is sufficiently smaller than the dimension of each mode $N_k$. Then, given error and confidence parameters $\epsilon, \delta \in (0, 1)$, there exists a constant $s = s(\epsilon, \delta)$ such that the approximated residual $\ell_{R_1, \ldots, R_K}(X|_{S_1, \ldots, S_K})$ is close to the original one $\ell_{R_1, \ldots, R_K}(X)$, to within $\epsilon$ with a probability of at least $1 - \delta$. Note that the time

complexity for computing $\ell_{R_1,\ldots,R_K}(X|_{S_1,\ldots,S_K})$ does not depend on the input size $N_1,\ldots,N_K$ but rather on the sample size $s$, meaning that the algorithm runs in *constant time*, regardless of the input size.

The main component in our proof is the weak version of Szemerédi's regularity lemma [9], which roughly states that any tensor can be well approximated by a tensor consisting of a constant number of blocks whose entries in the same block are equal. Then, we can show that $X|_{S_1,\ldots,S_K}$ is a good sketch of the original tensor, because by sampling $s$ many indices for each mode, we can hit each block a sufficient number of times. It follows that $\ell_{R_1,\ldots,R_K}(X)$ and $\ell_{R_1,\ldots,R_K}(X|_{S_1,\ldots,S_K})$ are close. To formalize this argument, we want to measure the "distance" between $X$ and $X|_{S_1,\ldots,S_K}$, and we want to show that it is small. To this end, we exploit graph limit theory, first described by Lovász and Szegedy [13] (see also [12]), in which we measure the distance between two graphs on a different number of vertices by considering continuous versions called *dikernels*. Hayashi and Yoshida [10] used graph limit theory to develop a constant-time algorithm that minimizes quadratic functions described by matrices and vectors. We further extend this theory to tensors to analyze the Tucker fitting problem.

With both synthetic and real datasets, we numerically evaluate our algorithm. The results show that our algorithm overwhelmingly outperforms other approximation methods in terms of both speed and accuracy.

## 2   Preliminaries

**Tensors**   Let $X \in \mathbb{R}^{N_1 \times \cdots N_K}$ be a tensor. Then, we define the *Frobenius norm* of $X$ as $\|X\|_F = \sqrt{\sum_{i_1,\ldots,i_K} X^2_{i_1\cdots i_K}}$, the *max norm* of $X$ as $\|X\|_{\max} = \max_{i_1 \in [N_1],\ldots,i_K \in [N_K]} |X_{i_1\cdots i_K}|$, and the *cut norm* of $X$ as $\|X\|_\square = \max_{S_1 \subseteq [N_1],\ldots,S_K \subseteq [N_K]} \left| \sum_{i_1 \in S_1,\ldots,i_K \in S_K} X_{i_1\cdots i_K} \right|$. We note that these norms satisfy the triangle inequalities.

For a vector $\boldsymbol{v} \in \mathbb{R}^n$ and a sequence $S = (x_1,\ldots,x_s)$ of indices in $[n]$, we define the *restriction* $\boldsymbol{v}|_S \in \mathbb{R}^s$ of $\boldsymbol{v}$ such that $(\boldsymbol{v}|_S)_i = v_{x_i}$ for $i \in [s]$. Let $X \in \mathbb{R}^{N_1 \times \cdots \times N_K}$ be a tensor, and $S_k = (x^k_1,\ldots,x^k_s)$ be a sequence of indices in $[N_k]$ for each mode $k \in [K]$. Then, we define the *restriction* $X|_{S_1,\ldots,S_K} \in \mathbb{R}^{s \times \cdots \times s}$ of $X$ to $S_1 \times \cdots \times S_K$ such that $(X|_{S_1,\ldots,S_K})_{i_1\cdots i_K} = X_{x^1_{i_1},\ldots,x^K_{i_K}}$ for each $i_1 \in [N_1],\ldots,i_K \in [N_k]$.

**Hyper-dikernels**   We call a (measurable) function $\mathcal{W} : [0,1]^K \to \mathbb{R}$ a *(hyper-)dikernel of order $K$*. We can regard a dikernel as a tensor whose indices are specified by real values in $[0,1]$. We stress that the term "dikernel" has nothing to do with kernel methods used in machine learning.

For two functions $f, g : [0,1] \to \mathbb{R}$, we define their inner product as $\langle f, g \rangle = \int_0^1 f(x)g(x)\mathrm{d}x$. For a sequence of functions $f^{(1)},\ldots,f^{(K)}$, we define their tensor product $\bigotimes_{k \in [K]} f^{(k)} \in [0,1]^K \to \mathbb{R}$ as $\bigotimes_{k \in [K]} f^{(k)}(x_1,\ldots,x_K) = \prod_{k \in [K]} f^{(k)}(x_k)$, which is an order-$K$ dikernel.

Let $\mathcal{W} : [0,1]^K \to \mathbb{R}$ be a dikernel. Then, we define the *Frobenius norm* of $\mathcal{W}$ as $\|\mathcal{W}\|_F = \sqrt{\int_{[0,1]^K} \mathcal{W}(\boldsymbol{x})^2 \mathrm{d}\boldsymbol{x}}$, the *max norm* of $\mathcal{W}$ as $\|\mathcal{W}\|_{\max} = \max_{\boldsymbol{x} \in [0,1]^K} |\mathcal{W}(\boldsymbol{x})|$, and the *cut norm* of $\mathcal{W}$ as $\|\mathcal{W}\|_\square = \sup_{S_1,\ldots,S_K \subseteq [0,1]} \left| \int_{S_1 \times \cdots \times S_K} \mathcal{W}(\boldsymbol{x})\mathrm{d}\boldsymbol{x} \right|$. Again, we note that these norms satisfy the triangle inequalities. For two dikernels $\mathcal{W}$ and $\mathcal{W}'$, we define their inner product as $\langle \mathcal{W}, \mathcal{W}' \rangle = \int_{[0,1]^K} \mathcal{W}(\boldsymbol{x})\mathcal{W}'(\boldsymbol{x})\mathrm{d}\boldsymbol{x}$.

Let $\lambda$ be a Lebesgue measure. A map $\pi : [0,1] \to [0,1]$ is said to be *measure-preserving*, if the pre-image $\pi^{-1}(X)$ is measurable for every measurable set $X$, and $\lambda(\pi^{-1}(X)) = \lambda(X)$. A *measure-preserving bijection* is a measure-preserving map whose inverse map exists and is also measurable (and, in turn, also measure-preserving). For a measure-preserving bijection $\pi : [0,1] \to [0,1]$ and a dikernel $\mathcal{W} : [0,1]^K \to \mathbb{R}$, we define a dikernel $\pi(\mathcal{W}) : [0,1]^K \to \mathbb{R}$ as $\pi(\mathcal{W})(x_1,\ldots,x_K) = \mathcal{W}(\pi(x_1),\ldots,\pi(x_K))$.

For a tensor $G \in \mathbb{R}^{R_1 \times \cdots \times R_K}$ and vector-valued functions $\{F^{(k)} : [0,1] \to \mathbb{R}^{R_k}\}_{k \in [K]}$, we define an order-$K$ dikernel $[\![G; F^{(1)}, \ldots, F^{(K)}]\!] : [0,1]^K \to \mathbb{R}$ as

$$[\![G; F^{(1)}, \ldots, F^{(K)}]\!](x_1, \ldots, x_K) = \sum_{r_1 \in [R_1], \ldots, r_K \in [R_K]} G_{r_1, \ldots, r_K} \prod_{k \in [K]} F^{(k)}(x_k)_{r_k}$$

We note that $[\![G; F^{(1)}, \ldots, F^{(K)}]\!]$ is a continuous analogue of Tucker decomposition.

**Tensors and hyper-dikernels**  We can construct the dikernel $\mathcal{X} : [0,1]^K \to \mathbb{R}$ from a tensor $X \in \mathbb{R}^{N_1 \times \cdots \times N_K}$ as follows. For an integer $n \in \mathbb{N}$, let $I_1^n = [0, \frac{1}{n}], I_2^n = (\frac{1}{n}, \frac{2}{n}], \ldots, I_n^n = (\frac{n-1}{n}, \ldots, 1]$. For $x \in [0,1]$, we define $i_n(x) \in [n]$ as a unique integer such that $x \in I_i^n$. Then, we define $\mathcal{X}(x_1, \ldots, x_K) = X_{i_{N_1}(x_1) \cdots i_{N_K}(x_K)}$. The main motivation for creating a dikernel from a tensor is that, in doing so, we can define the distance between two tensors $X$ and $Y$ of different sizes via the cut norm—that is, $\|\mathcal{X} - \mathcal{Y}\|_\square$.

Let $\mathcal{W} : [0,1]^K \to \mathbb{R}$ be a dikernel and $S_k = (x_1^k, \ldots, x_s^k)$ for $k \in [K]$ be sequences of elements in $[0,1]$. Then, we define a dikernel $\mathcal{W}|_{S_1, \ldots, S_K} : [0,1]^K \to \mathbb{R}$ as follows: We first extract a tensor $W \in \mathbb{R}^{s \times \cdots \times s}$ by setting $W_{i_1 \cdots i_K} = \mathcal{W}(x_{i_1}^1, \ldots, x_{i_K}^K)$. Then, we define $\mathcal{W}|_{S_1, \ldots, S_K}$ as the dikernel constructed from $W$.

## 3  Correctness of Algorithm 1

In this section, we prove the correctness of Algorithm 1.

The following sampling lemma states that dikernels and their sampling versions are close in the cut norm with high probability.

**Lemma 3.1.** *Let $\mathcal{W}^1, \ldots, \mathcal{W}^T : [0,1]^K \to [-L, L]$ be dikernels. Let $S_1, \ldots, S_K$ be sequences of $s$ elements uniformly and independently sampled from $[0,1]$. Then, with a probability of at least $1 - \exp(-\Omega_K(s^2 (T/\log_2 s)^{1/(K-1)}))$, there exists a measure-preserving bijection $\pi : [0,1] \to [0,1]$ such that, for every $t \in [T]$, we have*

$$\|\mathcal{W}^t - \pi(\mathcal{W}^t|_{S_1, \ldots, S_K})\|_\square = L \cdot O_K (T/\log_2 s)^{1/(2K-2)},$$

*where $O_K(\cdot)$ and $\Omega_K(\cdot)$ hide factors depending on $K$.*

We now consider the dikernel counterpart to the Tucker fitting problem, in which we want to compute the following:

$$\ell_{R_1, \ldots, R_K}(\mathcal{X}) := \inf_{G \in \mathbb{R}^{R_1 \times \cdots \times R_K}, \{f^{(k)} : [0,1] \to \mathbb{R}^{R_k}\}_{k \in [K]}} \left\| \mathcal{X} - [\![G; f^{(1)}, \ldots, f^{(K)}]\!] \right\|_F^2, \qquad (2)$$

The following lemma states that the Tucker fitting problem and its dikernel counterpart have the same optimum values.

**Lemma 3.2.** *Let $X \in \mathbb{R}^{N_1 \times \cdots \times N_K}$ be a tensor, and let $R_1, \ldots, R_K \in \mathbb{N}$ be integers. Then, we have*

$$\ell_{R_1, \ldots, R_K}(X) = \ell_{R_1, \ldots, R_K}(\mathcal{X}).$$

For a set of vector-valued functions $F = \{f^{(k)} : [0,1] \to \mathbb{R}^{R_k}\}_{k \in [K]}$, we define $\|F\|_{\max} = \max_{k \in [K], r \in [R_k], x \in [0,1]} f_r^{(k)}(x)$. For real values $a, b, c \in \mathbb{R}$, $a = b \pm c$ is shorthand for $b - c \leq a \leq b + c$. For a dikernel $\mathcal{X} : [0,1]^K \to \mathbb{R}$, we define a dikernel $\mathcal{X}^2 : [0,1]^K \to \mathbb{R}$ as $\mathcal{X}^2(\boldsymbol{x}) = \mathcal{X}(\boldsymbol{x})^2$ for every $\boldsymbol{x} \in [0,1]^K$. The following lemma states that if $\mathcal{X}$ and $\mathcal{Y}$ are close in the cut norm, then the optimum values of the Tucker fitting problem regarding them are also close.

**Lemma 3.3.** *Let $\mathcal{X}, \mathcal{Y} : [0,1]^K \to \mathbb{R}$ be dikernels with $\|\mathcal{X} - \mathcal{Y}\|_\square \leq \epsilon$ and $\|\mathcal{X}^2 - \mathcal{Y}^2\|_\square \leq \epsilon$. For integers $R_1, \ldots, R_K \in \mathbb{N}$, we have*

$$\ell_{R_1, \ldots, R_K}(\mathcal{X}) = \ell_{R_1, \ldots, R_K}(\mathcal{Y}) \pm 2\epsilon \Big( 1 + R \big( \|G_\mathcal{X}\|_{\max} \|F_\mathcal{X}\|_{\max}^K + \|G_\mathcal{Y}\|_{\max} \|F_\mathcal{Y}\|_{\max}^K \big) \Big),$$

*where $(G_\mathcal{X}, F_\mathcal{X} = \{f_\mathcal{X}^{(k)}\}_{k \in [K]})$ and $(G_\mathcal{Y}, F_\mathcal{Y} = \{f_\mathcal{Y}^{(k)}\}_{k \in [K]})$ are solutions to the problem (2) on $\mathcal{X}$ and $\mathcal{Y}$, respectively, whose objective values exceed the infima by at most $\epsilon$, and $R = \prod_{k \in [K]} R_k$.*

It is well known that the Tucker fitting problem has a minimizer for which the factor matrices are orthonormal. Thus, we have the following guarantee for the approximation error of Algorithm 1.

**Theorem 3.4.** *Let* $X \in \mathbb{R}^{N_1 \times \cdots \times N_K}$ *be a tensor,* $R_1, \ldots, R_K$ *be integers, and* $\epsilon, \delta \in (0, 1)$. *For* $s(\epsilon, \delta) = 2^{\Theta(1/\epsilon^{2K-2})} + \Theta(\log \frac{1}{\delta} \log \log \frac{1}{\delta})$, *we have the following. Let* $S_1, \ldots, S_K$ *be sequences of indices as defined in Algorithm 1. Let* $(G^*, U_1^*, \ldots, U_K^*)$ *and* $(\tilde{G}^*, \tilde{U}_1^*, \ldots, \tilde{U}_K^*)$ *be minimizers of the problem* (1) *on* $X$ *and* $X|_{S_1, \ldots, S_K}$ *for which the factor matrices are orthonormal, respectively. Then, with a probability of at least* $1 - \delta$, *we have*

$$\ell_{R_1, \ldots, R_K}(X|_{S_1, \ldots, S_K}) = \ell_{R_1, \ldots, R_K}(X) \pm O(\epsilon L^2(1 + 2MR)),$$

*where* $L = \|X\|_{\max}$, $M = \max\{\|G^*\|_{\max}, \|\tilde{G}^*\|_{\max}\}$, *and* $R = \prod_{k \in [K]} R_k$.

We remark that, for the matrix case (i.e., $K = 2$), $\|G^*\|_{\max}$ and $\|\tilde{G}^*\|_{\max}$ are equal to the maximum singular values of the original and sampled matrices, respectively.

*Proof.* We apply Lemma 3.1 to $\mathcal{X}$ and $\mathcal{X}^2$. Then, with a probability of at least $1 - \delta$, there exists a measure-preserving bijection $\pi : [0, 1] \to [0, 1]$ such that

$$\|\mathcal{X} - \pi(\mathcal{X}|_{S_1, \ldots, S_K})\|_\square \leq \epsilon L \quad \text{and} \quad \|\mathcal{X}^2 - \pi(\mathcal{X}^2|_{S_1, \ldots, S_K})\|_\square \leq \epsilon L^2.$$

In what follows, we assume that this has happened. Then, by Lemma 3.3 and the fact that $\ell_{R_1, \ldots, R_K}(\mathcal{X}|_{S_1, \ldots, S_K}) = \ell_{R_1, \ldots, R_K}(\pi(\mathcal{X}|_{S_1, \ldots, S_K}))$, we have

$$\ell_{R_1, \ldots, R_K}(\mathcal{X}|_{S_1, \ldots, S_K}) = \ell_{R_1, \ldots, R_K}(\mathcal{X}) \pm \epsilon L^2 \Big( 1 + 2R(\|G\|_{\max} \|F\|_{\max}^K + \|\tilde{G}\|_{\max} \|\tilde{F}\|_{\max}^K) \Big),$$

where $(G, F = \{f^{(k)}\}_{k \in [K]})$ and $(\tilde{G}, \tilde{F} = \{\tilde{f}^{(k)}\}_{k \in [K]})$ be as in the statement of Lemma 3.3. From the proof of Lemma 3.2, we can assume that $\|G\|_{\max} = \|G^*\|_{\max}$, $\|\tilde{G}\|_{\max} = \|\tilde{G}^*\|_{\max}$, $\|F\|_{\max} \leq 1$, and $\|\tilde{F}\|_{\max} \leq 1$ (owing to the orthonormality of $U_1^*, \ldots, U_K^*$ and $\tilde{U}_1^*, \ldots, \tilde{U}_K^*$). It follows that

$$\ell_{R_1, \ldots, R_K}(\mathcal{X}|_{S_1, \ldots, S_K}) = \ell_{R_1, \ldots, R_K}(\mathcal{X}) \pm \epsilon L^2 \Big( 1 + 2R(\|G^*\|_{\max} + \|\tilde{G}^*\|_{\max}) \Big). \tag{3}$$

Then, we have

$$\ell_{R_1, \ldots, R_K}(X|_{S_1, \ldots, S_K}) = \ell_{R_1, \ldots, R_K}(\mathcal{X}|_{S_1, \ldots, S_K}) \qquad \text{(By Lemma 3.2)}$$

$$= \ell_{R_1, \ldots, R_K}(\mathcal{X}) \pm \epsilon L^2 \Big( 1 + 2R(\|G^*\|_{\max} + \|\tilde{G}^*\|_{\max}) \Big) \qquad \text{(By (3))}$$

$$= \ell_{R_1, \ldots, R_K}(X) \pm \epsilon L^2 \Big( 1 + 2R(\|G^*\|_{\max} + \|\tilde{G}^*\|_{\max}) \Big). \qquad \text{(By Lemma 3.2)}$$

Hence, we obtain the desired result. $\qquad \square$

## 4 Related Work

To solve Tucker decomposition, several randomized algorithms have been proposed. A popular approach involves using a truncated or randomized SVD. For example, Zhou *et al.* [21] proposed a variant of HOOI with randomized SVD. Another approach is based on tensor sparsification. Tsourakakis [17] proposed MACH, which randomly picks the element of the input tensor and substitutes zero, with a probability of $1 - p$, where $p \in (0, 1]$ is an approximation parameter. Moreover, several authors proposed CUR-type Tucker decomposition, which approximates the input tensor by sampling tensor tubes [6, 8].

Unfortunately, these methods do not significantly reduce the computational cost. Randomized SVD approaches reduce the computational cost of multiple SVDs from $O(K \max_k N_k \cdot \prod_k N_k)$ to $O(K \max_k R_k \cdot \prod_k N_k)$, but they still depend on $\prod_k N_k$. CUR-type approaches require the same time complexity. In MACH, to obtain accurate results, we need to set $p$ as constant for instance $p = 0.1$ [17]. Although this will improve the runtime by a constant factor, the dependency on $\prod_k N_k$ does not change.

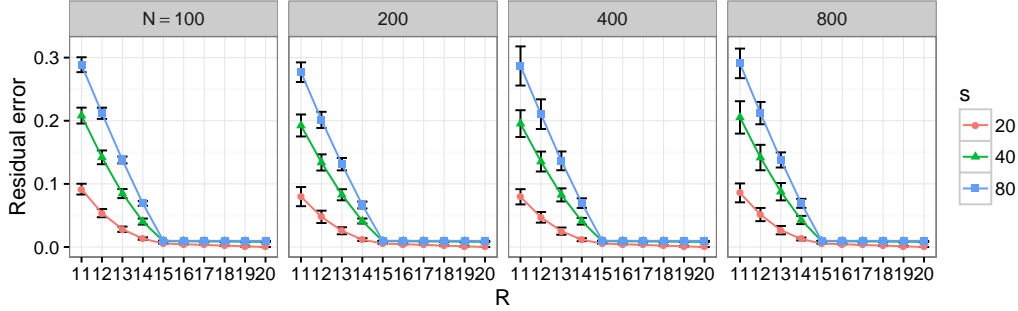

Figure 1: Synthetic data: computed residual errors for various Tucker ranks. The horizontal axis indicates the approximated residual error $\ell_{R_1,\ldots,R_K}(X|_{S_1,\ldots,S_K})$. The error bar indicates the standard deviation over ten trials with different random seeds, which affected both data generation and sampling.

## 5 Experiments

For the experimental evaluation, we slightly modified our sampling algorithm. In Algorithm 1, the indices are sampled using sampling with replacement (i.e., the same indices can be sampled more than once). Although this sampling method is theoretically sound, we risk obtaining redundant information by sampling the same index several times. To avoid this issue, we used sampling without replacement—i.e., each index was sampled at most once. Furthermore, if the dimension of a mode was smaller than the sampling size, we used all the coordinates. That is, we sampled $\min(s, N_k)$ indices for each mode $k \in [K]$. Note that both sampling methods, with and without replacement, are almost equivalent when the input size $N_1, \ldots, N_K$ is sufficiently larger than $s$ (i.e., the probability that a previously sampled index is sampled approaches zero.)

### 5.1 Synthetic Data

We first demonstrate the accuracy of our method using synthetic data. We prepared $N \times N \times N$ tensors for $N \in \{100, 200, 400, 800\}$, with a Tucker rank of $(15, 15, 15)$. Each element of the core $G \in \mathbb{R}^{15 \times 15 \times 15}$ and the factor matrices $U^{(1)}, U^{(2)}, U^{(3)} \in \mathbb{R}^{N \times 15}$ was drawn from a standard normal distribution. We set $Y = [\![G; U^{(1)}, U^{(2)}, U^{(3)}]\!]$. Then, we generated $X \in \mathbb{R}^{N \times N \times N}$ as $X_{ijk} = Y_{ijk}/\|Y\|_F + 0.1\epsilon_{ijk}$, where $\epsilon_{ijk}$ follows the standard normal distribution for $i, j, k \in [N]$. Namely, $X$ had a low-rank structure, though some small noise was added. Subsequently, $X$ was decomposed using our method with various Tucker ranks $(R, R, R)$ for $R \in \{11, 12, \ldots, 20\}$ and the sample size $s \in \{20, 40, 80\}$.

The results (see Figure 1) show that our method behaved ideally. That is, the error was high when $R$ was less than the true rank, 15, and it was almost zero when $R$ was greater than or equal to the true rank. Note that the scale of the estimated residual error seems to depend on $s$, i.e., small $s$ tends to yield a small residual error. This implies our method underestimates the residual error when $s$ is small.

### 5.2 Real Data

To evaluate how our method worked against real data tensors, we used eight datasets [1, 2, 4, 11, 14, 19] described in Table 1, where the "fluor" dataset is order-4 and the others are order-3 tensors. Details regarding the data are provided in the Supplementary material. Before the experiment, we normalized each data tensor by its norm $\|X\|_F$. To evaluate the approximation accuracy, we used HOOI implemented in Python by Nickel[3] as "true" residual error.[4] As baselines, we used the two randomized methods introduced in Section 4: randomized SVD [21] and MACH [17]. We denote our method by "sample$s$" where $s$ indicates the sample size (e.g., sample40 denotes our method with

Table 1: Real Datasets.

| Dataset | Size | Total # of elements |
|---|---|---|
| movie_gray | $120 \times 160 \times 107$ | 2.0M |
| EEM | $28 \times 13324 \times 8$ | 2.9M |
| fluorescence | $299 \times 301 \times 41$ | 3.6M |
| bonnie | $89 \times 97 \times 549$ | 4.7M |
| fluor | $405 \times 136 \times 19 \times 5$ | 5.2M |
| wine | $44 \times 2700 \times 200$ | 23.7M |
| BCI_Berlin | $4001 \times 59 \times 1400$ | 0.3G |
| visor | $16818 \times 288 \times 384$ | 1.8G |

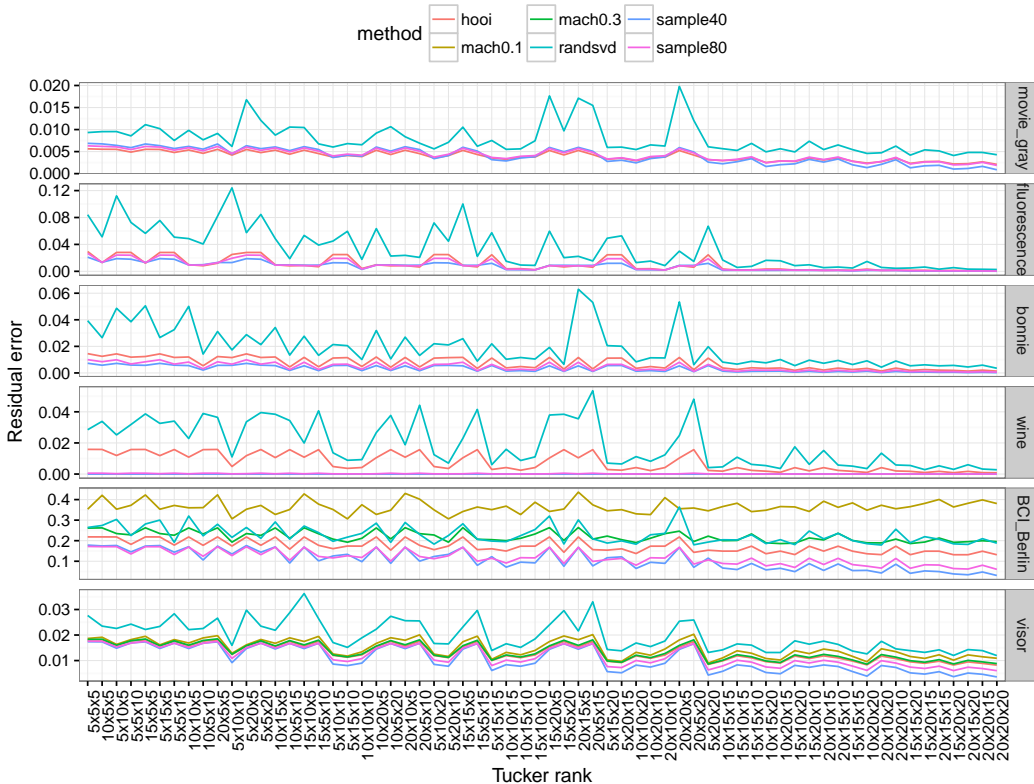

Figure 2: Real data: (approximated) residual errors for various Tucker ranks.

$s = 40$). Similarly, "mach$p$" refers to MACH with sparsification probability set at $p$. For all the approximation methods, we used the HOOI implementation to solve Tucker decomposition. Every data tensor was decomposed with Tucker rank $(R_1, \ldots, R_K)$ on the grid $R_k \in \{5, 10, 15, 20\}$ for $k \in [K]$.

Figure 2 shows the residual error for order-3 data.[5] It shows that the random projection tends to overestimate the decomposition error. On the other hand, except for the wine dataset, our method stably estimated the residual error with reasonable approximation errors. For the wine dataset, our method estimated a very small value, far from the correct value. This result makes sense, however, because the wine dataset is sparse (where $90\%$ of the elements are zero) and the residual error is too small. Table 2 shows the absolute error from HOOI averaged over all rank settings. In most of the datasets, our methods achieved the lowest error.

Table 2: Real data: absolute error of HOOI's and other's residual errors averaged over ranks. The best and the second best results are shown in bold and italic, respectively.

|  | mach0.1 | mach0.3 | randsvd | sample40 | sample80 |
|---|---|---|---|---|---|
| movie_gray | $0.084 \pm 0.038$ | $0.020 \pm 0.010$ | $0.004 \pm 0.003$ | *0.001 ± 0.001* | **0.000 ± 0.000** |
| EEM | $2.370 \pm 0.792$ | $0.587 \pm 0.210$ | $0.018 \pm 0.029$ | *0.003 ± 0.003* | **0.003 ± 0.003** |
| fluorescence | $0.569 \pm 0.204$ | $0.129 \pm 0.053$ | $0.024 \pm 0.023$ | *0.004 ± 0.005* | **0.002 ± 0.002** |
| bonnie | $1.170 \pm 0.412$ | $0.300 \pm 0.121$ | $0.012 \pm 0.011$ | *0.004 ± 0.002* | **0.003 ± 0.001** |
| fluor | $0.611 \pm 0.307$ | $0.148 \pm 0.083$ | $0.009 \pm 0.007$ | *0.003 ± 0.001* | **0.002 ± 0.001** |
| wine | $6.826 \pm 0.733$ | $1.417 \pm 0.191$ | $0.012 \pm 0.009$ | *0.008 ± 0.006* | **0.007 ± 0.006** |
| BCI_Berlin | $0.193 \pm 0.039$ | **0.048 ± 0.013** | $0.057 \pm 0.020$ | $0.065 \pm 0.022$ | *0.055 ± 0.007* |
| visor | $0.002 \pm 0.001$ | **0.000 ± 0.000** | $0.007 \pm 0.003$ | $0.003 \pm 0.001$ | *0.001 ± 0.001* |

Table 3: Real data: Kendall's tau against the ranking of Tucker ranks obtained by HOOI.

|  | mach0.1 | mach0.3 | randsvd | sample40 | sample80 |
|---|---|---|---|---|---|
| movie_gray | -0.07 | 0.04 | 0.1 | *0.71* | **0.73** |
| EEM | 0.64 | 0.68 | 0.77 | *0.79* | **0.91** |
| fluorescence | 0.08 | 0.02 | 0.28 | *0.61* | **0.77** |
| bonnie | -0.05 | -0.01 | *0.33* | 0.27 | **0.67** |
| fluor | 0.77 | 0.73 | 0.83 | **0.93** | *0.89* |
| wine | *0.12* | 0.12 | -0.02 | 0.04 | **0.15** |
| BCI_Berlin | 0.08 | 0.09 | 0.02 | *0.18* | **0.45** |
| visor | 0.07 | 0.18 | 0.11 | *0.64* | **0.7** |

Table 4: Real data: runtime averaged over Tucker ranks (in seconds).

|  | hooi | mach0.1 | mach0.3 | randsvd | sample40 | sample80 |
|---|---|---|---|---|---|---|
| movie_gray | 0.71 | 32.19 | 85.13 | 0.33 | **0.13** | *0.25* |
| EEM | 3447.97 | 7424.8 | 7938.75 | 2212.54 | **0.11** | *0.11* |
| fluorescence | 2.67 | 30.05 | 73.52 | 1.47 | **0.13** | *0.23* |
| bonnie | 9.13 | 25.99 | 56.56 | 2.32 | **0.11** | *0.41* |
| fluor | 3.2 | 34.54 | 98.63 | 1.43 | **0.2** | *0.43* |
| wine | 142.34 | 95.28 | 212.19 | 41.94 | **0.12** | *0.23* |
| BCI_Berlin | 428.13 | 2765.88 | 7830 | 82.43 | **0.2** | *0.45* |
| visor | 10034.96 | 27897.85 | 27769.53 | 1950.45 | **0.13** | *0.26* |

Next, we evaluated the correctness of the order of Tucker ranks. For rank determination, it is important that the rankings of Tucker ranks in terms of residual errors are consistent between the original and the sampled tensors. For example, if the rank-$(15, 15, 5)$ Tucker decomposition of the original tensor achieves a lower error than the rank-$(5, 15, 15)$ Tucker decomposition, this order relation should be preserved in the sampled tensor. We evaluated this using Kendall's tau coefficient, between the rankings of Tucker ranks obtained by HOOI and the others. Kendall's tau coefficient takes as its value $+1$ when the two rankings are the same, and $-1$ when they are opposite. Table 3 shows the results. We can see that, again, our method outperformed the others.

Table 4 shows the runtime averaged over all the rank settings. It shows that our method is consistently the fastest. Note that MACH was slower than normal Tucker decomposition. This is possibly because it must create an additional sparse tensor, which requires $O(\prod_k N_k)$ time complexity.

## 6 Discussion

One might point out by way of criticism that the residual error is not a satisfying measure for determining rank. In machine learning and statistics, it is common to choose hyperparameters based on the generalization error or its estimator, such as cross-validation (CV) error, rather than the training error (i.e., the residual error in Tucker decomposition). Unfortunately, our approach cannot be used the CV error, because what we can obtain is the minimum of the training error, whereas CV requires us to plug in the minimizers. An alternative is to use information criteria such as Akaike [3] and Bayesian information criteria [15]. These criteria are given by the penalty term, which consists of

the number of parameters and samples[6], and the maximum log-likelihood. Because the maximum log-likelihood is equivalent to the residual error, our method can approximate these criteria.

Python code of our algorithm is available at: https://github.com/hayasick/CTFT.

## Footnotes

[3] https://github.com/mnick/scikit-tensor

[4] Note that, though no approximation is used in HOOI, the objective function (1) is nonconvex and it is not guaranteed to converge to the global minimum. The obtained solution can be different from the ground truth.

[5]Here we exclude the results of the EEM dataset because its size is too small and we were unable to run the experiment with all the Tucker rank settings. Also, the results of MACH on some datasets are excluded owing to considerable errors.

[6]For models with multiple solutions, such as Tucker decomposition, the penalty term can differ from the standard form [20]. Still, these criteria are useful in practice (see, e.g. [16]).

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
