[Supplementary Material · supplementary.pdf]

# Supplementary Material for "Fitting Low-Rank Tensors in Constant Time"

## A    Proof of Lemma 3.1

Before proving Lemma 3.1, we need to establish several definitions. We say that a partition $\mathcal{Q}$ is a *refinement* of another partition $\mathcal{P} = (V_1, \ldots, V_p)$ if $\mathcal{Q}$ is obtained by splitting each set $V_i$ into one or more parts. The partition $\mathcal{P} = (V_1, \ldots, V_p)$ of the interval $[0,1]$ is called an *equipartition* if $\lambda(V_i) = 1/p$ for every $i \in [p]$. For a dikernel $\mathcal{W} : [0,1]^K \to \mathbb{R}$ and an equipartition $\mathcal{P} = (V_1, \ldots, V_p)$ of $[0,1]$, we define $\mathcal{W}_{\mathcal{P}} : [0,1]^K \to \mathbb{R}$ as the dikernel obtained by averaging each $V_{i_1} \times \cdots \times V_{i_K}$ for $i_1, \ldots, i_K \in [p]$. More formally, we define

$$\mathcal{W}_{\mathcal{P}}(\boldsymbol{x}) = \frac{1}{\prod_{k \in [K]} \lambda(V_{i_k})} \int_{V_{i_1} \times \cdots \times V_{i_k}} \mathcal{W}(\boldsymbol{x}') \mathrm{d}\boldsymbol{x}' = p^K \int_{V_{i_1} \times \cdots \times V_{i_K}} \mathcal{W}(\boldsymbol{x}') \mathrm{d}\boldsymbol{x}',$$

where $i_k$ is the unique index such that $x_k \in V_{i_k}$ for each $k \in [K]$.

The following lemma states that any dikernel $\mathcal{W} : [0,1]^K \to \mathbb{R}$ can be well approximated by $\mathcal{W}_{\mathcal{P}}$ for an equipartition $\mathcal{P}$ into a small number of parts.

**Lemma A.1** (Weak regularity lemma for dikernels [9])**.** *Let $\mathcal{P}$ be an equipartition of $[0,1]$ into $p$ sets. Then, for any dikernel $\mathcal{W} : [0,1]^K \to \mathbb{R}$ and $\epsilon > 0$, there exists a refinement $\mathcal{Q}$ of $\mathcal{P}$ with $|\mathcal{Q}| \le p2^{O(1/\epsilon^{2K-2})}$ such that*

$$\|\mathcal{W} - \mathcal{W}_{\mathcal{Q}}\|_{\square} \le \epsilon \|\mathcal{W}\|_F.$$

**Corollary A.2.** *Let $\mathcal{W}^1, \ldots, \mathcal{W}^T : [0,1]^K \to \mathbb{R}$ be dikernels. Then, for any $\epsilon > 0$, there exists an equipartition $\mathcal{P}$ into $|\mathcal{P}| \le 2^{O(T/\epsilon^{2K-2})}$ parts, such that for every $t \in [T]$,*

$$\|\mathcal{W}^t - \mathcal{W}^t_{\mathcal{P}}\|_{\square} \le \epsilon \|\mathcal{W}^t\|_F.$$

*Proof.* Let $\mathcal{P}^0$ be a trivial partition, that is, a partition consisting of a single part $[0,1]$. Then, for each $t \in [T]$, we iteratively apply Lemma A.1 with $\mathcal{P}^{t-1}$, $\mathcal{W}^t$, and $\epsilon$, and we obtain the partition $\mathcal{P}^t$ into at most $|\mathcal{P}^{t-1}|2^{O(1/\epsilon^{2K-2})}$ parts such that $\|\mathcal{W}^t - \mathcal{W}^t_{\mathcal{P}^t}\|_{\square} \le \epsilon \|\mathcal{W}^t\|_2$. Because $\mathcal{P}^t$ is a refinement of $\mathcal{P}^{t-1}$, we have $\|\mathcal{W}^i - \mathcal{W}^i_{\mathcal{P}^t}\|_{\square} \le \|\mathcal{W}^i - \mathcal{W}^i_{\mathcal{P}^{t-1}}\|_{\square}$ for every $i \in [t-1]$. Then, $\mathcal{P}^T$ satisfies the desired property with $|\mathcal{P}^T| \le \left(2^{O(1/\epsilon^{2K-2})}\right)^T = 2^{O(T/\epsilon^{2K-2})}$. $\qquad\qquad\square$

Although the following lemma was originally proved for order-2 dikernels, the proof can easily be extended to general orders:

**Lemma A.3** ((4.15) of [5])**.** *Let $\mathcal{W} : [0,1]^K \to [-L, L]$ be a dikernel, and let $S_1, \ldots, S_K$ be sequences of $s$ elements uniformly and independently sampled from $[0,1]$. Then, we have*

$$-\frac{L}{s^{\Omega_K(1)}} \le \mathop{\mathbf{E}}_{S_1, \ldots, S_K} \|\mathcal{W}|_{S_1, \ldots, S_K}\|_{\square} - \|\mathcal{W}\|_{\square} < \frac{L}{s^{\Omega_K(1)}},$$

*where $\Omega_K(1)$ hides a factor depending on $K$.*

Finally, we need the following concentration inequality.

**Lemma A.4** (Azuma's inequality)**.** *Let $(\Omega, A, P)$ be a probability space, $k$ be a positive integer, and $C > 0$. Let $\boldsymbol{z} = (z_1, \ldots, z_k)$, where $z_1, \ldots, z_k$ are independent random variables, and $z_i$ takes values in some measure space $(\Omega_i, A_i)$. Let $f : \Omega_1 \times \cdots \times \Omega_k \to \mathbb{R}$ be a function. Suppose that $|f(\boldsymbol{x}) - f(\boldsymbol{y})| \le C$ whenever $\boldsymbol{x}$ and $\boldsymbol{y}$ only differ in one coordinate. Then*

$$\Pr\left[|f(\boldsymbol{z}) - \mathop{\mathbf{E}}_{\boldsymbol{z}}[f(\boldsymbol{z})]| > \lambda C\right] < 2e^{-\lambda^2/2k}.$$

*Proof of Lemma 3.1.* We first bound the expectations and then prove their concentrations. We apply Corollary A.2 to $\mathcal{W}^1, \dots, \mathcal{W}^T$ and $\epsilon$, and let $\mathcal{P} = (V_1, \dots, V_p)$ be the obtained partition with $p \leq 2^{T/\epsilon^{2K-2}}$ parts such that

$$\|\mathcal{W}^t - \mathcal{W}^t_{\mathcal{P}}\|_{\square} \leq \epsilon L.$$

for every $t \in [T]$. According to Lemma A.3, for every $t \in [T]$, we have

$$\underset{S_1,\dots,S_K}{\mathbf{E}} \|\mathcal{W}^t_{\mathcal{P}}|_{S_1,\dots,S_K} - \mathcal{W}^t|_{S_1,\dots,S_K}\|_{\square} = \underset{S_1,\dots,S_K}{\mathbf{E}} \|(\mathcal{W}^t_{\mathcal{P}} - \mathcal{W}^t)|_{S_1,\dots,S_K}\|_{\square} \leq \epsilon L + \frac{L}{s^{\Omega_K(1)}}.$$

Then, for any measure-preserving bijection $\pi : [0,1] \to [0,1]$ and $t \in [T]$, we have

$$\underset{S_1,\dots,S_K}{\mathbf{E}} \|\mathcal{W}^t - \pi(\mathcal{W}^t|_{S_1,\dots,S_K})\|_{\square} \tag{4}$$

$$\leq \|\mathcal{W}^t - \mathcal{W}^t_{\mathcal{P}}\|_{\square} + \underset{S_1,\dots,S_K}{\mathbf{E}} \|\mathcal{W}^t_{\mathcal{P}} - \pi(\mathcal{W}^t_{\mathcal{P}}|_{S_1,\dots,S_K})\|_{\square} + \underset{S_1,\dots,S_K}{\mathbf{E}} \|\pi(\mathcal{W}^t_{\mathcal{P}}|_{S_1,\dots,S_K}) - \pi(\mathcal{W}^t|_{S_1,\dots,S_K})\|_{\square}$$

$$\leq 2\epsilon L + \frac{L}{s^{\Omega_K(1)}} + \underset{S_1,\dots,S_K}{\mathbf{E}} \|W^t_{\mathcal{P}} - \pi(W^t_{\mathcal{P}}|_{S_1,\dots,S_K})\|_{\square}. \tag{5}$$

Thus, we are left with the problem of sampling from $\mathcal{P}$. For each $k \in [K]$, let $Z_i^k$ be the number of points in $S_k$ that fall into the set $V_i$. It is easy to compute the following:

$$\mathbf{E}[Z_i^k] = \frac{s}{p} \quad \text{and} \quad \mathbf{Var}[Z_i^k] = \left(\frac{1}{p} - \frac{1}{p^2}\right)s < \frac{s}{p}$$

for every $k \in [K]$. For each $k \in [K]$, the partition $\mathcal{P}^k$ of $[0,1]$ into the sets $V_1^k, \dots, V_p^k$ is constructed such that $\lambda(V_i^k) = Z_i^k/s$ and $\lambda(V_i \cap V_i^k) = \min(1/p, Z_i^k/s)$. For each $t \in [T]$, we construct the dikernel $\overline{\mathcal{W}}^t : [0,1]^K \to \mathbb{R}$ such that the value of $\overline{\mathcal{W}}^t$ on $V_{i_1}^1 \times \cdots V_{i_K}^K$ is the same as the value of $\mathcal{W}^t_{\mathcal{P}}$ on $V_{i_1}^1 \times \cdots \times V_{i_K}^K$. Then, $\overline{\mathcal{W}}^t$ agrees with $\mathcal{W}^t_{\mathcal{P}}$ on the set $Q = \bigcup_{i_1,\dots,i_K \in [p]} (V_{i_1} \cap V_{i_1}^1) \times \cdots \times (V_{i_K} \cap V_{i_K}^K)$. Then, there exists a bijection $\pi$ such that $\pi(\mathcal{W}^t_{\mathcal{P}}|_{S_1,\dots,S_K}) = \overline{\mathcal{W}}^t$ for each $t \in [T]$. Then, for every $t \in [T]$, we have

$$\|\mathcal{W}^t_{\mathcal{P}} - \pi(\mathcal{W}^t_{\mathcal{P}}|_{S_1,\dots,S_K})\|_{\square} = \|\mathcal{W}^t_{\mathcal{P}} - \overline{\mathcal{W}}^t\|_{\square} \leq \int |\mathcal{W}^t_{\mathcal{P}}(\boldsymbol{x}) - \overline{\mathcal{W}}^t(\boldsymbol{x})| \mathrm{d}\boldsymbol{x} \leq 2L(1 - \lambda(Q))$$

$$= 2L\left(1 - \prod_{k \in [K]} \sum_{i \in [p]} \min\left(\frac{1}{p}, \frac{Z_i^k}{s}\right)\right) = 2L\left(1 - \prod_{k \in [K]} \left(1 - \frac{1}{2}\sum_{i \in [p]} \left|\frac{1}{p} - \frac{Z_i^k}{s}\right|\right)\right)$$

$$\leq 2L\left(1 - \prod_{k \in [K]} \left(1 - \frac{\sqrt{p}}{2}\sqrt{\sum_{i \in [p]} \left(\frac{1}{p} - \frac{Z_i^k}{s}\right)^2}\right)\right)$$

$$\leq L\sqrt{p} \sum_{k \in [K]} \sqrt{\sum_{i \in [p]} \left(\frac{1}{p} - \frac{Z_i^k}{s}\right)^2}.$$

Then, we have

$$\mathbf{E}\|\mathcal{W}^t_{\mathcal{P}} - \pi(\mathcal{W}^t_{\mathcal{P}}|_{S_1,\dots,S_K})\|_{\square} = L\sqrt{p} \sum_{k \in [K]} \mathbf{E}\sqrt{\sum_{i \in [p]} \left(\frac{1}{p} - \frac{Z_i^k}{s}\right)^2} \leq L\sqrt{p} \sum_{k \in [K]} \sqrt{\mathbf{E}\sum_{i \in [p]} \left(\frac{1}{p} - \frac{Z_i^k}{s}\right)^2}$$

$$\leq L\sqrt{p} \sum_{k \in [K]} \sqrt{\frac{1}{s^2}\sum_{i \in [p]} \mathbf{Var} Z_i^k} \leq L\sqrt{p} \sum_{k \in [K]} \sqrt{\frac{1}{ps}} = KL\sqrt{\frac{p}{s}}.$$

Inserted this into (5), we obtain

$$\mathbf{E}\|\mathcal{W}^t - \pi(\mathcal{W}^t|_{S_1,\dots,S_K})\|_{\square} \leq 2\epsilon L + \frac{L}{s^{\Omega_K(1)}} + KL\sqrt{\frac{p}{s}} \leq 2\epsilon L + \frac{L}{s^{\Omega_K(1)}} + \frac{KL}{\sqrt{s}}2^{O(T/\epsilon^{2K-2})}.$$

Choosing $\epsilon = O\big(T/(\log_2 s^{\Omega_K(1)})\big)^{1/(2K-2)} = O_K\big(T/\log_2 s\big)^{1/(2K-2)}$, we obtain the upper bound

$$\mathbf{E}\,\|\mathcal{W}^t - \pi(\mathcal{W}^t|_{S_1,\dots,S_K})\|_\square \leq 2L \cdot O_K\Big(\frac{T}{\log_2 s}\Big)^{1/(2K-2)} + \frac{L}{s^{\Omega_K(1)}} + \frac{KL}{s^{\Omega_K(1)}} = L \cdot O_K\Big(\frac{T}{\log_2 s}\Big)^{1/(2K-2)}.$$

Observing that $\|\mathcal{W}^t - \pi(\mathcal{W}^t|_{S_1,\dots,S_K})\|_\square$ changes by at most $O(L/s)$ if an element in one of $S_1,\dots,S_K$ changes, we apply Azuma's inequality with $\lambda = s \cdot \Omega_K(T/\log_2 s)^{1/(2K-2)}$ and the union bound to complete the proof. $\qquad\square$

## B  Proof of Lemma 3.2

We say that a vector-valued function $f : [0,1] \to \mathbb{R}^R$ is *orthonormal* if $\langle f_r, f_r \rangle = 1$ for every $r \in [R]$ and $\langle f_r, f_{r'} \rangle = 0$ if $r \neq r'$. First, we calculate the partial derivatives of the objective function:

**Lemma B.1.** *Let $\mathcal{X} \in [0,1]^K \to \mathbb{R}$ be a dikernel, $G \in \mathbb{R}^{R_1 \times \cdots R_K}$ be a tensor, and $\{f^{(k)} : [0,1] \to \mathbb{R}^{R_k}\}_{k \in [K]}$ be a set of orthonormal vector-valued functions. Then, we have*

$$\frac{\partial}{\partial f_{r_0}^{(k_0)}(x_0)}\Big\|\mathcal{X} - [\![G; f^{(1)},\dots,f^{(K)}]\!]\Big\|_F^2$$

$$= 2 \sum_{r_1,\dots,r_K:r_{k_0}=r_0} G_{r_1\cdots r_K} \int_{[0,1]^K:x_{k_0}=x_0} \mathcal{X}(\boldsymbol{x}) \prod_{k\in[K]\backslash\{k_0\}} f_{r_k}^{(k)}(x_k)\mathrm{d}\boldsymbol{x}$$

$$-2\sum_{r_1,\dots,r_K} G_{r_1\cdots r_K} G_{r_1\cdots r_{k_0-1}r_0 r_{k_0+1}\cdots r_K} f_{r_{k_0}}^{(k_0)}(x_0).$$

*Proof.*

$$\frac{\partial}{\partial f_{r_0}^{(k_0)}(x_0)}\Big\|\mathcal{X} - [\![G; f^{(1)},\dots,f^{(K)}]\!]\Big\|_F^2 = \frac{\partial}{\partial f_{r_0}^{(k_0)}(x_0)} \int_{[0,1]^K}\Big(\mathcal{X}(\boldsymbol{x}) - \sum_{r_1,\dots,r_K} G_{r_1\cdots r_K}\prod_{k\in[K]} f_{r_k}^{(k)}(x_k)\Big)^2 \mathrm{d}\boldsymbol{x}$$

$$= 2\int_{[0,1]^K:x_{k_0}=x_0}\Big(\mathcal{X}(\boldsymbol{x}) - \sum_{r_1,\dots,r_K} G_{r_1\cdots r_K}\prod_{k\in[K]} f_{r_k}^{(k)}(x_k)\Big)\sum_{r_1',\dots,r_K':r_{k_0}'=r_0} G_{r_1'\cdots r_K'}\prod_{k\in[K]\backslash\{k_0\}} f_{r_k'}^{(k)}(x_k)\mathrm{d}\boldsymbol{x}$$

$$= 2\sum_{r_1,\dots,r_K:r_{k_0}=r_0} G_{r_1\cdots r_K}\int_{[0,1]^K:x_{k_0}=x_0} \mathcal{X}(\boldsymbol{x})\prod_{k\in[K]\backslash\{k_0\}} f_{r_k}^{(k)}(x_k)\mathrm{d}\boldsymbol{x}$$

$$-2\sum_{r_1,\dots,r_K} G_{r_1\cdots r_K}\sum_{r_1',\dots,r_K':r_{k_0}'=r_0} G_{r_1'\cdots r_K'} f_{r_{k_0}}^{(k_0)}(x_0)\int_{[0,1]^K:x_{k_0}=x_0}\prod_{k\in[K]\backslash\{k_0\}} f_{r_k}^{(k)}(x_k)\prod_{k\in[K]\backslash\{k_0\}} f_{r_k'}^{(k)}(x_k)\mathrm{d}\boldsymbol{x}$$

$$= 2\sum_{r_1,\dots,r_K:r_{k_0}=r_0} G_{r_1\cdots r_K}\int_{[0,1]^K:x_{k_0}=x_0} \mathcal{X}(\boldsymbol{x})\prod_{k\in[K]\backslash\{k_0\}} f_{r_k}^{(k)}(x_k)\mathrm{d}\boldsymbol{x}$$

$$-2\sum_{r_1,\dots,r_K} G_{r_1\cdots r_K}\sum_{r_1',\dots,r_K':r_{k_0}'=r_0} G_{r_1'\cdots r_K'} f_{r_{k_0}}^{(k_0)}(x_0)\prod_{k\in[K]\backslash\{k_0\}}\int_{[0,1]} f_{r_k}^{(k)}(x_k) f_{r_k'}^{(k)}(x_k)\mathrm{d}x_k$$

$$= 2\sum_{r_1,\dots,r_K:r_{k_0}=r_0} G_{r_1\cdots r_K}\int_{[0,1]^K:x_{k_0}=x_0} \mathcal{X}(\boldsymbol{x})\prod_{k\in[K]\backslash\{k_0\}} f_{r_k}^{(k)}(x_k)\mathrm{d}\boldsymbol{x}$$

$$-2\sum_{r_1,\dots,r_K} G_{r_1\cdots r_K} G_{r_1\cdots r_{k_0-1}r_0 r_{k_0+1}\cdots r_K} f_{r_{k_0}}^{(k_0)}(x_0).$$

which completes the proof. $\qquad\square$

*Proof of Lemma 3.2.* First, we show that (LHS) $\leq$ (RHS). Consider a sequence of solutions for the continuous problem (2) whose objective values attains the infimum. For Tucker decompositions, it is well known that there exists a minimizer for which the factor matrices $U^{(1)},\dots,U^{(K)}$ are orthonormal. By a similar reasoning, we can show that the vector-valued functions $f^{(1)},\dots,f^{(K)}$ in each solution of the sequence are orthonormal. As the objective function is coercive with respect

to the tensor $G$, we can take a subsequence for which $G$ converges. Let $G^*$ be the limit. Now, for any $\delta > 0$, we can create a matrix $\tilde{G}$ by perturbing $G^*$ so that (i) by fixing $G$ to $\tilde{G}$ in the continuous problem, the infimum increases only by $\delta$, and (ii) a matrix constructed from $\tilde{G}$ is invertible (the detail is given later) and has a condition number at least $\delta' = \delta'(\delta)$.

Now, consider a sequence of solutions for the continuous problem (2) with $G$ fixed to $\tilde{G}$ whose objective values attains the infimum. We can show that the partial derivatives converges to zeros almost everywhere. Then, for any $\epsilon > 0$, there exists a solution $(\tilde{G}, f^{(1)}, \dots, f^{(K)})$ in the sequence such that the partial derivatives are at most $\epsilon$ almost everywhere.

Then by Lemma B.1, for any $k_0 \in [K]$, $r_0 \in [R_k]$, and almost all $x \in [0,1]$, we have

$$\sum_{r_1, \dots, r_K} \tilde{G}_{r_1 \cdots r_K} \tilde{G}_{r_1 \cdots r_{k_0-1} r_0 r_{k_0+1} \cdots r_K} f^{(k_0)}_{r_{k_0}}(x_0) \tag{6}$$

$$= \sum_{r_1, \dots, r_K : r_{k_0} = r_0} \tilde{G}_{r_1 \cdots r_K} \int_{[0,1]^K : x_{k_0} = x_0} \mathcal{X}(\boldsymbol{x}) \prod_{k \in [K] \setminus \{k_0\}} f^{(k)}_{r_k}(x_k) \mathrm{d}\boldsymbol{x} \pm \epsilon(k_0, r_0, x), \tag{7}$$

where $\epsilon(k_0, r_0, x) = O(\epsilon)$. Now, we consider a system of linear equations consisting of (7) for $r_0 = 1, \dots, r_{k_0}$. We assume that the matrix involved in this system is invertible and has a condition number at least $\delta'$. Then, for any $k, r \in [R_k]$ and almost every pair $x, x' \in [0,1]$ with $i_{N_k}(x) = i_{N_k}(x')$, we have $f^{(k_0)}_{r_0}(x) = f^{(k_0)}_{r_0}(x') \pm O(\epsilon/\delta')$. For each $k \in [K]$, we can define a matrix $U^{(k)} \in \mathbb{R}^{N_k \times R_k}$ as $U^{(k)}_{ir} = f^{(k)}_r(x)$, where $x \in [0,1]$ is an arbitrary value with $i_{N_k}(x) = i$. Then, we have

$$\frac{1}{N} \left\| X - [\![\tilde{G}; U^{(1)}, \dots, U^{(K)}]\!] \right\|_F^2 = \frac{1}{N} \sum_{i_1, \dots, i_K} \left( X_{i_1 \cdots i_K} - [\![\tilde{G}; U^{(1)}, \dots, U^{(K)}]\!]_{i_1 \cdots i_K} \right)^2$$

$$= \sum_{i_1, \dots, i_K} \int_{I_{i_1}^{N_1} \times \cdots \times I_{i_K}^{N_K}} \left( \mathcal{X}(\boldsymbol{x}) - [\![\tilde{G}; f^{(1)}, \dots, f^{(K)}]\!](\boldsymbol{x}) \pm O(\epsilon/\delta') \right)^2 \mathrm{d}\boldsymbol{x}$$

$$= \left\| \mathcal{X} - [\![\tilde{G}; f^{(1)}, \dots, f^{(K)}]\!] \right\|_F^2 \pm O(\epsilon^2 N / (\delta')^2)$$

for $N = \prod_{k \in [K]} N_k$. As the choice of $\epsilon$ and $\delta$ are arbitrary, we obtain (LHS) $\leq$ (RHS).

Second, we show that (RHS) $\leq$ (LHS). Let $U^{(k)} \in \mathbb{R}^{N_k \times R_k}$ ($k \in [K]$) be matrices. We define a vector-valued function $f^{(k)} : [0,1] \to \mathbb{R}^{R_k}$ as $f^{(k)}_r(x) = U^{(k)}_{i_{N_k}(x)r}$ for each $k \in [K]$ and $r \in [R_k]$. Then, we have

$$\left\| \mathcal{X} - [\![G; f^{(1)}, \dots, f^{(K)}]\!] \right\|_F^2 = \int_{[0,1]^K} \left( \mathcal{X}(\boldsymbol{x}) - [\![G; f^{(1)}, \dots, f^{(K)}]\!](\boldsymbol{x}) \right)^2 \mathrm{d}\boldsymbol{x}$$

$$= \sum_{i_1, \dots, i_K} \int_{\prod_{k \in [K]} I_{i_k}^{N_k}} \left( \mathcal{X}(\boldsymbol{x}) - [\![G; f^{(1)}, \dots, f^{(K)}]\!](\boldsymbol{x}) \right)^2 \mathrm{d}\boldsymbol{x}$$

$$= \frac{1}{N} \sum_{i_1, \dots, i_K} \left( X_{i_1 \cdots i_K} - [\![G; U^{(1)}, \dots, U^{(K)}]\!]_{i_1 \cdots i_K} \right)^2$$

$$= \frac{1}{N} \left\| X - [\![G; U^{(1)}, \dots, U^{(K)}]\!] \right\|_F^2. \qquad \square$$

## C   Proof of Lemma 3.3

The cut norm is useful to bound the absolute value of the inner product between a tensor and a tensor product:

**Lemma C.1.** *Let $\epsilon \geq 0$ and $\mathcal{W} : [0,1]^K \to \mathbb{R}$ be a dikernel with $\|\mathcal{W}\|_\square \leq \epsilon$. Then, for any functions $f^{(1)}, \dots, f^{(K)} : [0,1] \to [-L, L]$, we have $|\langle \mathcal{W}, \bigotimes_{k \in [K]} f^{(k)} \rangle| \leq \epsilon L^K$.*

*Proof.* For $\tau \in \mathbb{R}$ and the function $h : [0,1] \to \mathbb{R}$, let $L_\tau(h) := \{x \in [0,1] \mid h(x) = \tau\}$ be the level set of $h$ at $\tau$. For $f'^{(i)} = f^{(i)}/L$, we have

$$\left|\langle \mathcal{W}, \bigotimes_{k \in [K]} f^{(k)}\rangle\right| = L^K |\langle \mathcal{W}, \bigotimes_{k \in [K]} f'^{(k)}\rangle| = L^K \left|\int_{[-1,1]^K} \prod_{k \in [K]} \tau_k \int_{\prod_{k \in [K]} L_{\tau_k}(f'^{(k)})} \mathcal{W}(\boldsymbol{x}) \mathrm{d}\boldsymbol{x} \mathrm{d}\boldsymbol{\tau}\right|$$

$$\leq L^K \int_{[-1,1]^K} \prod_{k \in [K]} |\tau_k| \left|\int_{\prod_{k \in [K]} L_{\tau_k}(f'^{(k)})} \mathcal{W}(\boldsymbol{x}) \mathrm{d}\boldsymbol{x} \mathrm{d}\boldsymbol{\tau}\right| \leq \epsilon L^K \int_{[-1,1]^K} \prod_{k \in [K]} |\tau_k| \mathrm{d}\boldsymbol{\tau} = \epsilon L^K. \quad \square$$

Thus, we have the following:

**Lemma C.2.** *Let $\mathcal{X}, \mathcal{Y} : [0,1]^K \to \mathbb{R}$ be dikernels with $\|\mathcal{X} - \mathcal{Y}\|_\square \leq \epsilon$ and $\|\mathcal{X}^2 - \mathcal{Y}^2\|_\square \leq \epsilon$, where $\mathcal{X}^2(\boldsymbol{x}) = \mathcal{X}(\boldsymbol{x})^2$ and $\mathcal{Y}^2(\boldsymbol{x}) = \mathcal{Y}(\boldsymbol{x})^2$ for every $\boldsymbol{x} \in [0,1]^K$. Then, for any tensor $G \in \mathbb{R}^{R_1 \times \cdots \times R_K}$ and a set of vector-valued functions $F = \{f^{(k)} : [0,1] \to \mathbb{R}^{R_k}\}_{k \in [K]}$, we have*

$$\left\|\mathcal{X} - [\![G; f^{(1)}, \ldots, f^{(K)}]\!]\right\|_F^2 = \left\|\mathcal{Y} - [\![G; f^{(1)}, \ldots, f^{(K)}]\!]\right\|_F^2 \pm \epsilon\left(1 + 2R\|G\|_{\max}\|F\|_{\max}^K\right),$$

*where $R = \prod_{k \in [K]} R_K$.*

*Proof.* We have

$$\left|\left\|\mathcal{X} - [\![G; f^{(1)}, \ldots, f^{(K)}]\!]\right\|_F^2 - \left\|\mathcal{Y} - [\![G; f^{(1)}, \ldots, f^{(K)}]\!]\right\|_F^2\right|$$

$$= \left|\int_{[0,1]^K} \left(\mathcal{X}(\boldsymbol{x}) - [\![G; f^{(1)}, \ldots, f^{(K)}]\!](\boldsymbol{x})\right)^2 \mathrm{d}\boldsymbol{x} - \int_{[0,1]^K} \left(\mathcal{Y}(\boldsymbol{x}) - [\![G; f^{(1)}, \ldots, f^{(K)}]\!](\boldsymbol{x})\right)^2 \mathrm{d}\boldsymbol{x}\right|$$

$$= \left|\int_{[0,1]^K} \left(\mathcal{X}(\boldsymbol{x})^2 - \mathcal{Y}(\boldsymbol{x})^2\right)\mathrm{d}\boldsymbol{x} - 2\int_{[0,1]^K} (\mathcal{X}(\boldsymbol{x}) - \mathcal{Y}(\boldsymbol{x}))[\![G; f^{(1)}, \ldots, f^{(K)}]\!](\boldsymbol{x})\mathrm{d}\boldsymbol{x}\right|$$

$$\leq \|\mathcal{X}^2 - \mathcal{Y}^2\|_\square + 2 \sum_{r_1 \in [R_1], \ldots, r_k \in [R_k]} |G_{r_1 \cdots r_K}| \cdot \left|\langle \mathcal{X} - \mathcal{Y}, \bigotimes_{k \in [K]} f_{r_k}^{(k)}\rangle\right|$$

$$\leq \epsilon + 2\epsilon R\|G\|_{\max}\|F\|_{\max}^K$$

by Lemma C.1. $\quad \square$

*Proof of Lemma 3.3.* By Lemma C.2, we have

$$\left\|\mathcal{Y} - [\![G_{\mathcal{Y}}; f_{\mathcal{Y}}^{(1)}, \ldots, f_{\mathcal{Y}}^{(K)}]\!]\right\|_F^2 \leq \left\|\mathcal{Y} - [\![G_{\mathcal{X}}; f_{\mathcal{X}}^{(1)}, \ldots, f_{\mathcal{X}}^{(K)}]\!]\right\|_F^2 + \epsilon$$

$$\leq \left\|\mathcal{X} - [\![G_{\mathcal{X}}; f_{\mathcal{X}}^{(1)}, \ldots, f_{\mathcal{X}}^{(K)}]\!]\right\|_F^2 + \left(2\epsilon + 2\epsilon R\|G_{\mathcal{X}}\|_{\max}\|F_{\mathcal{X}}\|_{\max}^K\right).$$

Similarly, we have

$$\left\|\mathcal{X} - [\![G_{\mathcal{X}}; f_{\mathcal{X}}^{(1)}, \ldots, f_{\mathcal{X}}^{(K)}]\!]\right\|_F^2 \leq \left\|\mathcal{X} - [\![G_{\mathcal{Y}}; f_{\mathcal{Y}}^{(1)}, \ldots, f_{\mathcal{Y}}^{(K)}]\!]\right\|_F^2 + \epsilon$$

$$\leq \left\|\mathcal{Y} - [\![G_{\mathcal{Y}}; f_{\mathcal{Y}}^{(1)}, \ldots, f_{\mathcal{Y}}^{(K)}]\!]\right\|_F^2 + \left(2\epsilon + 2\epsilon R\|G_{\mathcal{Y}}\|_{\max}\|F_{\mathcal{Y}}\|_{\max}^K\right).$$

Hence, the claim follows. $\quad \square$

## D  Description of real datasets

**movie_gray:** One of the movies contained in a human activity video dataset [14]. It consists of 107 frames at $120 \times 160$ resolution. The original movie had RGB color information, but we reduce it to monochrome.

**EEM:** A collection of samples measured using fluorescence spectroscopy forming Excitation-Emission Matrices (EEMs)[7] [2].

**fluorescence:** A collection of EEM measurements of human blood plasma samples[8] [11]. We used the variable `X_UD` in the dataset.

**bonnie:** HPLC-PDA profiles of 24 commercial preparations of St. John's wort, originating from several continents[9] [1].

**fluor:** A fluorescence dataset[10].

**wine:** 3-way data contained in the Wine GC-MS FT-IR dataset[11].

**BCI_Berlin:** Generated from electroencephalogram (EEG) data collected in Berlin[12] [4]. It records EEG signals with 59 channels for multiple trials and subjects. A signal is recorded each millisecond. From matlab files `BCICIV_calib_ds1a_1000Hz.mat`, `BCICIV_calib_ds1b_1000Hz.mat`, ..., `BCICIV_calib_ds1g_1000Hz.mat`, we extracted 4001 frames from each trial start-point and then concatenated them.

**visor:** Generated from video surveillance data[13] [19]. We extracted each frame of the video and converted it to a monochrome image. There were 16,818 frames at $288 \times 384$ resolution.

## Footnotes

[7] http://www.models.life.ku.dk/joda/prototype

[8] http://www.models.life.ku.dk/anders-cancer

[9] http://www.models.life.ku.dk/Bonnie

[10] http://www.models.life.ku.dk/Fluorescence

[11] http://www.models.life.ku.dk/Wine_GCMS_FTIR

[12] http://www.bbci.de/competition/iv/desc_1.html

[13] http://www.openvisor.org/video_details.asp?idvideo=285