[Reviews · NeurIPS 2017]

Reviewer 1



Motivated by the problem of selecting an appropriate Tucker rank for low-rank tensor approximation, the authors provide an algorithm for estimating the RMSE of a tensor at a given Tucker rank. The algorithm is simple: randomly sample a subset of the indices in each mode, restrict to the resulting tensor, and compute the RMSE for the given Tucker rank of this smaller tensor. Their theory argues that the number of samples needed to get a good additive error approximation to the RMSE at a given Tucker rank depends on intrinsic parameters of the problem rather than the dimensions of the tensor: in particular, on the order of the tensor, the largest magnitude entry in the tensor (its spikiness), the spikiness of the Tucker core tensor of the original tensor, and the spikiness of the Tucker core tensor of the sampled tensor. The experimental results are impressive, and show that not only does this algorithm provide good estimates of the RMSE at given Tucker ranks (compared to the HOOI baseline), it also preserves the ordering of the Tucker ranks by the RMSEs this achieve. The latter point is crucial for model selection. This work is interesting and worth publishing in NIPS because it tackles the problem of rank selection for low-rank tensor approximation in a novel way, provides theoretical guarantees that seem reasonable, and the resulting algorithm is both fast and does a good job of rank selection. The main drawback of this paper is that the theory is presented in a non-intuitive manner that makes it hard to understand the connection that the authors draw between tensors and dikernels and how it is useful in establishing the main result (I did not read the supplementary material). - I would like to see the authors address the dependence on the spikiness of the Tucker core tensor of the subsampled tensor: is there a reason this should be here, or can it be removed? can this spikiness be bounded in terms of properties of the original tensor, or can it be arbitrarily poorly behaved? - It would be useful to state your theorem 3.4 for a matrix first, so the readers can use their experience in that domain to understand the quantities involved in your bounds - The argument in line 162 that Lemma 3.3 implies the following equation is not clear to me, since it seems to ignore the effect of the measure-preserving bijection. I did not check the supplementary material to see if this argument is made clearer, but if so, please state this in the paper. Another point: - I would like to see bold and italicization in Tables 2--4 to indicate the best and second best performing methods, as is standard

Reviewer 2



The paper proposes to approximate the minimum value of the Tucker tensor decomposition optimization problem (as this is directly related to Tucker rank). The algorithm the paper proposed is very efficient. The problem is indeed interesting and useful. The theoretical contribution is indeed novel and is based on the weak version of Szemerédi’s regularity lemma. Experiments demonstrate the practical usefulness of the algorithm. Summary. I believe that the algorithm is novel and the theoretical expositions interesting, useful and important. I believe that this a very good paper to be presented in NIPS. I believe that the paper should be accepted in the main programme.

Reviewer 3



Summary: This paper introduces an approximation algorithm 'sample_s' to approximate the original tensor X using an a constant small-sized sub-tensor X_tilt, which is 'extracted' from the original via the random sampling of sub-indices, such that the residual error of Tucker decomposition of X_tilt can stay close to that of X. And the algorithm depends only on the sampling size 's', which is free of the original tensor size 'N'. Strength: 1. By adapting the concept of 'dikernels' (and the associated theoretical result) to tensor representation, this paper provides theoretical analysis that characterizes the bounds of the residual error of tensor for the proposed algorithm. 2. The paper is clearly written, both the motivation and proposed solution can be easily grasped from the beginning of the text. Weakness: 1. The real applications that the proposed method can be applied to seem to be rather restricted. It seems the proposed algorithm can only be used as a fast evaluation of residual error for 'guessing' or 'predetermining' the range of Tucker ranks, not the real ranks. 2. Since the sampling size 's' depends on the exponential term 2^[1/(e^2K-2)] (in Theorem 3.4), it could be very large if one requires the error tolerance 'e' to be relatively small and the order of tensor 'K' to be high. In that situation, there won't be much benefits to use this algorithm. Question: 1. In the Fig.1, why blue curve with large sample size 's=80' achieves the worst error compared with that of red curve with small sample size 's=20'? Overall, although proposed algorithm is theoretically sound, but appears be limited in applications for practical propose.